# The Impact of Diabetes on Male Silkworm Reproductive Health

**DOI:** 10.3390/biology13080557

**Published:** 2024-07-24

**Authors:** Xiaoyan Zeng, Li Tong

**Affiliations:** 1Qinghai University, Xining 810000, China; zhcxbdy@163.com; 2Qinghai Provincial Key Laboratory of Traditional Chinese Medicine Research for Glucolipid Metabolic Diseases, Xining 810000, China

**Keywords:** diabetic reproductive damage model, silkworm, LH, T, GSH-Px, SOD, MDA, siwi1, sowi2

## Abstract

**Simple Summary:**

Diabetic male reproductive damage is a prevalent complication of type II diabetes, prompting a multitude of experimental investigations. However, the use of mammalian models for studying diabetic reproductive damage is expensive and raises ethical concerns. Hence, the identification of alternative animal models is imperative. The scientific community is increasingly recognizing the use of silkworms *Bombyx mori* as animal models, primarily because of their numerous advantages, including low cost and fewer ethical concerns. This study established a type II diabetic silkworm model via a high-glucose diet. The impact of diabetes on the reproductive system of male silkworms was observed, confirming the potential for using silkworms to create a model for diabetic reproductive damage.

**Abstract:**

The increasing prevalence of diabetic reproductive complications has prompted the development of innovative animal models. The use of the silkworm *Bombyx mori* as a model for diabetic reproductive damage shows potential as a valuable research tool. This study employed silkworms as a novel model to investigate diabetic reproductive damage. The silkworms were fed a high-glucose diet containing 10% glucose to induce a diabetic model. Subsequently, the study concentrated on assessing the influence of diabetes on the reproductive system of male silkworms. The results indicate that diabetes resulted in reduced luteinizing hormone (LH) and testosterone (T) levels, as well as elevated triglyceride (TG) levels in male silkworms. Moreover, diabetes mellitus was associated with pathological testicular damage in male silkworms, accompanied by decreased glutathione peroxidase (GSH-Px) and superoxide dismutase (SOD) levels, along with increased malondialdehyde (MDA) levels in the testis. Additionally, diabetes mellitus reduced the expression of siwi1 and siwi2 genes in the testis of male silkworms. Overall, these results support using silkworms as a valuable model for studying diabetic reproductive damage.

## 1. Introduction

Diabetes is a chronic metabolic disease characterized by hyperglycemia [1]. The International Diabetes Federation (IDF) has reported a rapid global increase in diabetes cases, projected to reach 783 million by 2045, with type II diabetes representing over 90% of these cases [2]. Type II diabetes has the potential to predispose individuals to a range of conditions, including cardiovascular diseases, peripheral neuropathy, and damage to the reproductive system [3]. Prolonged hyperglycemia can harm the male reproductive system by affecting testicular seminiferous tubules, interstitial cells, spermatogenic cells, and other tissues, resulting in reduced semen quality and increased male infertility rates [3]. Current therapeutic interventions for diabetes, encompassing insulin injections and oral hypoglycemic agents, have the capacity to regulate blood glucose levels and partially ameliorate reproductive function [4]. Some traditional herbal extracts are also used in the treatment of common diseases such as diabetes [5,6,7]. However, these improvements may not be sufficient. Therefore, an urgent requirement exists for the advancement of more efficacious pharmaceuticals or treatments aimed at tackling male reproductive complications arising from diabetes.

The current animal models utilized in diabetes drug discovery and development predominantly encompass mammals like rats, mice, rabbits, and piglets [8,9]. However, the utilization of mammals for diabetes research presents drawbacks including high costs (time and labor costs) and ethical concerns related to animal welfare [10,11]. Thus, seeking novel animal species to supplement the currently used mammals to prepare animal models of diabetes is not only an entirely new challenge, but also saves limited medical resources. This approach is in accordance with the 3R principle of laboratory animal welfare, advocating for using lower-order animals or alternative methods in place of vertebrate experiments to achieve similar research goals.

In 2002, the International Society of Invertebrates officially recognized the silkworm as a model insect [12]. The silkworm possesses numerous natural variations, a short life cycle, moderate size, easy sexual dimorphism, a high reproductive rate, and easily extractable organs for in vitro experimentation. With the completion of the sequencing of the silkworm genome [13], the availability of silkworm bioinformatic resources, analysis platforms, and tools for functional gene identification have improved significantly. This progress positions the silkworm as a promising model organism for researchers exploring the pathogenesis of human diseases and developing novel pharmaceuticals.

Zhang et al.’s investigation revealed that the insulin-like peptide encoded by the silkworm gene shares approximately 40% similarity with human insulin [14]. Furthermore, the investigation identified 25 silkworm genes that directly corresponded to human genes associated with diabetes mellitus [14]. Matsumoto et al. successfully established a type II diabetes model using silkworms [4]. The researchers evaluated the hypoglycemic effects of pioglitazone and metformin in this model [4]. Certain scholars in China have likewise induced hyperlipidemia symptoms in silkworms through the administration of exogenous hormones [15]. Additionally, a study conducted in Spain demonstrated that feeding silkworms a combination of glucose and chow led to an increase in hemolymph glucose concentration and inhibited growth, and they successfully replicated symptoms of type II diabetes [16]. These studies illustrate the potential utility of silkworms as a model for type II diabetes and for exploring treatment modalities. Nevertheless, research on the effects of diabetes on the reproductive system of male silkworms is lacking.

Therefore, this study aimed to develop a silkworm type II diabetes model to study how diabetes impacts male silkworms’ reproductive health. The goal was to create a cost-effective model that can be used to screen for potential drugs to treat diabetic reproductive damage in the future.

## 2. Materials and Methods

### 2.1. Animals and Reagents

The silkworms were sourced from the Sericulture Research Institute, Chinese Academy of Agricultural Sciences. The animals used in this study were silkworms (invertebrates), with no existing ethical issues; thus ethical approval was not required. T, LH, and FSH enzyme-linked Immunoassay (ELISA) kits were purchased from Wuhan Eliot Biotechnology Co., Ltd. (Wuhan, China). SOD, MDA, and GSH-Px kits were purchased from Wuhan Eliot Biotechnology Co., Ltd. (Wuhan, China). The TG detection kit was purchased from Nanjing Jiancheng Bioengineering Institute (Nanjing, China). Hematoxylin was purchased from Wuhan Sevier Biotechnology Co., Ltd. (Wuhan, China). Eosin was purchased from Zhuhai Beso Biotechnology Co., Ltd. (Zhuhai, China). RNA Trizol Reagent was purchased from Hefei Bomei Biotechnology Co., Ltd. (Hefei, China). Isopropyl alcohol was purchased from Shanghai Sangong Biological Engineering Technology Service Co., Ltd. (Shanghai, China). Chloroform was purchased from Baori Physical Technologies Co., Ltd. (Baori, China). The protease inhibitor Cocktail and phenylthiourea (analytical grade) were purchased from the Sigma Chemical Company (Sigma-Aldrich, St. Louis, MO, USA).

### 2.2. Experimental Model 

A total of 60 fourth-instar male larvae were randomly chosen and acclimated for 3–5 days. Fifth-instar larvae weighing 0.90 g–1.00 g on the day of molting were selected for experimentation. The silkworms were randomly divided into two groups: a control group and a model group, each consisting of 30 silkworms. The two groups of silkworms were fed under standard temperature and humidity rearing conditions: a temperature of 26–28 °C and a relative humidity between 75% and 85%. Silkworm larvae in the control group were fed 20 g of artificial diet, Silkmate 2S (Nosan Co., Yokohama, Japan), every 12 h. Silkworms in the model group were given a 20 g high-glucose diet containing 10% glucose [4] every 12 h. The high-glucose diet was formulated by blending Silkmate 2S and D-glucose, with a ratio of 10:1. Glucose levels in hemolymph were measured immediately after 72 h of feeding. Hemolymph samples were collected by cutting the first proleg of the silkworms [17], and blood glucose levels were measured using a glucometer (Lifescan, Johnson & Johnson, Shanghai, China) and test strips. Upon contact with the hemolymph produced post-incision, the glucometer test strips were filled via capillary action. The dilution of the hemolymph sample is unnecessary. Before commencing the study, calibration solution was used to check the glucometer. Blood glucose levels greater than 11.1 mmol/L in the model group indicated successful modeling. Subsequently, 10 silkworms from each group were randomly selected to observe damage to the reproductive system. To investigate the impact of diabetes on the reproductive system of male silkworms, this study measured the concentrations of T, LH, and follicle-stimulating hormone (FSH) in the hemolymph, TG in the fat body tissue, and the levels of SOD, MDA, and GSH-Px in the testicular tissue. Furthermore, histopathological changes in the testicular tissue were visualized through hematoxylin and eosin (HE) staining, while the expression levels of siwi1 and siwi2 genes involved in the piRNA signaling pathway were assessed using real-time PCR. Thus, this study aimed to determine the feasibility of modeling diabetic-induced reproductive damage using silkworms.

### 2.3. Preparation of Silkworm Hemolymph

The first prolegs of silkworms were cut, and the hemolymph was collected by applying gentle pressure to the abdomen into a centrifuge tube placed in an ice bath. A small quantity of protease inhibitor cocktail and phenylthiourea (analytical grade) were added, followed by centrifugation at 10,000 rpm for 10 min [18] to eliminate the hematocrit. The resulting supernatant was then preserved at −80 °C in a refrigerator.

### 2.4. Preparation of Silkworm Fat Body Samples

The fat body was isolated from the dorsolateral region of each larva, washed with insect saline (10 mM Tris/HCl, 130 mM NaCl, 5 mM KCl, and 1 mM CaCl_2_). Subsequently, it was homogenized in 100 μL of PBS and 1 μL of a protease inhibitor cocktail, then incubated at 60 °C for 15 min, and finally centrifuged at 3000 r/min for 10 min at 4 °C. The resulting supernatant was collected as the fat body tissue lysate.

### 2.5. Determination of T, LH, FSH, and TG

The expression levels of T, LH, and FSH were determined by enzyme-linked immunoassay. The assay was conducted according to the instructions provided with the kit. Every ELISA kit underwent supplementary in-laboratory validation procedures. Briefly, recombinant protein standards at diverse concentrations were provided by the ELISA manufacturer and then introduced into a normal control sample. After deducting the analyte concentration of the unspiked control from the retrieved values in the spiked samples, we established a strong correlation between the spiked and recovery concentrations falling within the standard curve for each analyte.

A volume of 2.5 μL of fat body tissue lysate was added to 200 μL of TG detection reagent, thoroughly mixed, and then incubated at 37 °C for 5 min. Subsequently, the absorbance was measured at 570 nm using a microplate reader (Shenzhen Radiometer Life Sciences Co., Ltd., Shenzhen, China).

### 2.6. Histological Analysis of the Silkworm Testicles

Testicular tissues were isolated from the silkworms, fixed immediately in 4% paraformaldehyde, processed serially, and then embedded in paraffin. Following this, 3 μm tissue sections were prepared using a rotary slicer (Leica 2016, Wetzlar, Germany), and the deparaffinized tissue sections were stained with hematoxylin and eosin. Finally, a digital slide scanner (Shenzhen Shengqiang Technology Co., Ltd., Shenzhen, China) was utilized for the image acquisition of the sections, and specific areas were selected for capturing images at 100× and 400× magnifications to observe the particular lesion.

### 2.7. Determination of SOD, MDA, and GSH-Px

The testicular tissue was homogenized in cold PBS using a tissue homogenizer (Tiangen Biochemical Technology Co., Ltd., Beijing, China), followed by centrifugation at 2000× *g* and 4 °C for 10 min to collect the supernatant of the tissue homogenate. The levels of SOD, MDA, and GSH-Px were quantified following the protocols specified in the corresponding kit instructions.

### 2.8. Real-Time PCR

Primers were designed and synthesized by Shanghai Shengong Bioengineering Service Co., Ltd. (Shanghai, China) (Table 1). RNA was extracted from silkworm testes using the Trizol method, and the concentration and purity of the RNA were subsequently assessed (OD260/OD280 = 1.91). Reverse transcription was conducted following the provided instructions. The resulting cDNA was amplified under specific conditions (37 °C, 15 min; 85 °C, 5 s; and 4 °C, 60 min) and then diluted with ddH_2_O. The PCR reaction mixture was prepared, and the PCR program with online amplification detection was established following the provided instructions. The mRNA expression was determined by 2^−ΔΔCT^. 

### 2.9. Statistical Analysis

The data in this experiment were statistically analyzed using SPSS 23.0. Comparisons between groups were performed using Student’s *t*-test (significance *p* < 0.05).

## 3. Results

### 3.1. Type II Diabetes Silkworm Model

In mammals, type II diabetes mellitus is frequently concomitant with dyslipidemia. Aberrant glucose metabolism may result in decreased insulin utilization, heightened lipolytic hormone activity, and elevated blood lipid levels [19]. The results indicate that after 72 h of high-glucose diet feeding, silkworms in the model group had elevated blood glucose levels compared to the control group (*p* < 0.05) (Figure 1A). TG levels in the model group were elevated compared to the control group (*p* < 0.05) (Figure 1B). Our findings indicate that changes in blood glucose and lipid levels in silkworms closely resemble those seen in mammals, supporting the successful establishment of a silkworm model for type II diabetes.

### 3.2. Diabetic Silkworm Reproductive Damage Model

To explore the potential for modeling diabetic reproductive damage using silkworms, we analyzed alterations in the reproductive systems of male silkworms.

#### 3.2.1. FSH, LH, and T Levels in Diabetic Silkworms

In mammals and other higher animals, sex hormones such as FSH, LH, and T play a critical role in male growth, development, and reproduction. Diabetes can affect endocrine levels, resulting in alterations in T, FSH, and LH in males. Under chronic hyperglycemic conditions, LH production is inhibited and the basal secretion of T is diminished [20]. Currently, there is a paucity of research on the presence of similar sex hormones in silkworms compared to mammals and whether these hormones regulate silkworm reproduction. Thus, we conducted a study examining the levels of FSH, LH, and T in the hemolymph of silkworms to determine the presence of sex hormones in silkworms. The study revealed a decrease in the LH and T levels of silkworms in the model group compared to the control group (*p* < 0.05) (Figure 2B,C). Additionally, the FSH levels in the two groups showed no significant difference (Figure 2A). These results imply the presence of FSH, LH, and T in silkworms, and suggest that hyperglycemic conditions could potentially lower LH and T levels in silkworms.

#### 3.2.2. SOD, MDA, and GSH-Px Levels in the Testis of Diabetic Silkworms

Oxidative stress is identified as a crucial mechanism of diabetic testicular injury in mammals. Prolonged states of hyperglycemia and hyperlipidemia are known to suppress the activity of antioxidant enzymes, leading to oxidative stress and the subsequent impairment of cellular and tissue function. The SOD and GSH-Px levels in the model group were lower than the control group (*p* < 0.05) (Figure 3A,C), while the MDA levels were higher (*p* < 0.05) (Figure 3B). These findings indicate that diabetes can potentially induce oxidative stress in silkworm testis tissues.

#### 3.2.3. Diabetic Silkworm Testis Histopathological Changes

In the control group, intact testicular ectodermal cell structure was observed, along with numerous spermatogonia and intact seminal vesicles containing sperm bundles (Figure 4). Conversely, the model group manifested delayed testicular development, necrotic spermatogonia, seminal vesicles, and plasma fibrous exudation (Figure 4). These results unveil the potential of silkworms as a viable paradigm for investigating reproductive impairments due to diabetes.

#### 3.2.4. Diabetic Silkworm Testicular Tissue siwi1 and siwi2 mRNA Levels

The piRNA pathway exhibits a close association with reproductive functions in mammals [21]. In this study, we utilized RT-PCR to detect the mRNA levels of siwi1 and siwi2 in the testicular tissue of the silkworm. Our findings reveal a decrease in siwi1 and siwi2 mRNA levels in the model group compared to the control group (*p* < 0.05) (Figure 5A,B). These results prove that diabetes can impact the reproductive function of silkworms through the piRNA pathway.

## 4. Discussion

Animal models play a crucial role in studying human diseases, enabling researchers to scrutinize pathophysiological changes and illuminate the mechanisms of diseases. Furthermore, these models aid in screening potential drugs for efficacy [8]. Over the years, various invertebrate models have been proposed as alternatives to using mammals. This is due to ethical and regulatory issues, along with the high costs and specialized facilities required for animal maintenance [22,23]. Silkworms are considered a valuable model organism in life sciences due to their genetic characteristics and genomic size. With 28 pairs of chromosomes and approximately 450 million base pairs, their genome is significant, making them comparable to humans in genetic complexity [24]. This indicates the potential to conduct research and experiments using silkworms to understand various biological processes. In addition, the silkworms’ metabolic function of sugar resembles that of mammals, underscoring their relevance in studying metabolic pathways and diseases [25]. In this context, the silkworm has been identified as a potential model for studying diabetes mellitus [4]. At present, a standardized silkworm model for the investigation of diabetes-related complications has not been developed. In this study, silkworms were utilized as an animal model to establish a diabetic reproductive damage model. This methodology is straightforward and cost-effective, and it furnishes a suitable animal model for the assessment of treatments concerning diabetic reproductive damage and drug-screening applications.

Diabetes mellitus is linked to a range of complications that have the potential to induce harm to multiple systems, including male reproductive dysfunction [26]. Prolonged hyperglycemia can induce a reduction in LH levels and T secretion, subsequently contributing to compromised sperm maturation and diminished sperm production, culminating in male infertility [27,28,29]. Elevated lipids are a common complication of type II diabetes [30], and increased lipid levels can impact sperm quality in males [31]. Chronic hyperglycemia can result in pathological damage to testicular tissue [32]. SOD and GSH-Px are vital enzymes that scavenge reactive oxygen species (ROS). GSH-Px helps reduce lipid peroxidation, whereas SOD serves as an essential antioxidant enzyme by scavenging oxygen radicals. Under conditions of hyperglycemia, elevated ROS levels in testicular tissues may lead to the decreased activity of SOD and GSH-Px, along with increased lipid peroxidation [33]. This decrease in antioxidant enzyme activity can reduce antioxidant capacity and elevate MDA content in testicular tissue, ultimately resulting in testicular damage. 

The present study successfully established a type II diabetes model in male silkworms and examined its impact on their reproductive system. The findings reveal a notable elevation in blood glucose and lipids in type II diabetic silkworms. These diabetic silkworms exhibited reduced levels of T and LH, along with severe damage to testicular tissues. Moreover, there was a reduction in the activities of SOD and GSH-Px within the testes, whereas the MDA content showed an increase. These outcomes closely resemble the reproductive damage seen in humans with type II diabetes. These results align with experimental observations in mammalian models of diabetic reproductive impairment. Kong ZL et al. utilized rats to establish a model of diabetic reproductive injury, illustrating that diabetes triggered oxidative stress in rats, leading to hypogonadism and modified testicular morphology [34]. Chen Y et al.’s [35] investigation revealed analogous outcomes where persistent hyperglycemia induced oxidative stress in mouse testes. HE staining unveiled notable pathological alterations such as seminiferous tubule detachment or atrophy, germ cell degeneration, and structural constriction and detachment within the seminiferous tubules. Overall, this experiment suggests that the silkworm could be a valuable animal model for studying diabetic reproductive damage.

Reproductive function in invertebrates and mammals is linked to the piRNA signaling pathway. In the process of spermatogenesis, small non-coding RNA (sncRNAs) are involved in the regulation of gene expression in germ cells [36]. In germ cells, piRNA distinguishes itself from other sncRNAs by virtue of its cellular specificity and quantity. PiRNA forms a complex with Piwi proteins, which belong to the Argonaute protein family, resulting in the formation of a piRNA silencing complex [37,38,39,40]. Abnormalities in the piRNA pathway can result in impaired spermatogenesis, potentially leading to male infertility. The piRNA pathway plays a crucial role in spermatogenesis through a multitude of mechanisms, including transposon repression, translational regulation, germline stem cell maintenance, RNA degradation, gene defense, and pseudogene production. In the silkworm, piRNA is predominantly found in transposons and is likely to play a role in the development of silkworm germ cells by regulating transposon activity. Based on the silkworm genome database and known Piwi protein subfamily genes, two Piwi protein genes, siwi1 and siwi2, were identified through homology searches. These genes are prominently expressed in the testis and ovary of the silkworm [41]. The study of piRNA is essential for comprehending germ cells, reproductive functions, and genetic inheritance. Our research demonstrated a decrease in siwi1 and siwi2 mRNA levels in the testes of diabetic silkworms, which is consistent with findings in mammalian models. Therefore, using silkworms as a model organism to investigate diabetic-induced reproductive damage offers advantages over the lengthy and expensive experimental processes involved with mammalian studies.

## 5. Conclusions

We created a model for diabetic reproductive damage by subjecting silkworms to a high-glucose diet and conducted a series of experiments to confirm its validity. Our findings suggest that diabetes may potentially result in damage to the reproductive system of male silkworms: (1) diabetes may lead to a reduction in LH and T levels; (2) diabetes may induce pathological harm to the testis; (3) diabetes may result in a decrease in testicular GSH-Px and SOD levels, along with an increase in MDA levels; (4) diabetes may decrease the expression of siwi1 and siwi2 in the testis. This innovative method offers a reliable animal model for future studies on the development of diabetic reproductive damage and the identification of potential treatment options. This model is cost-effective and highly efficient for studying disease progression and assessing drug efficacy. Unlike traditional models, this approach tackles ethical animal welfare issues and lowers research expenses. We anticipate this model will be pivotal in future studies focused on preventing and managing reproductive complications in individuals with diabetes.

## Figures and Tables

**Figure 1 biology-13-00557-f001:**
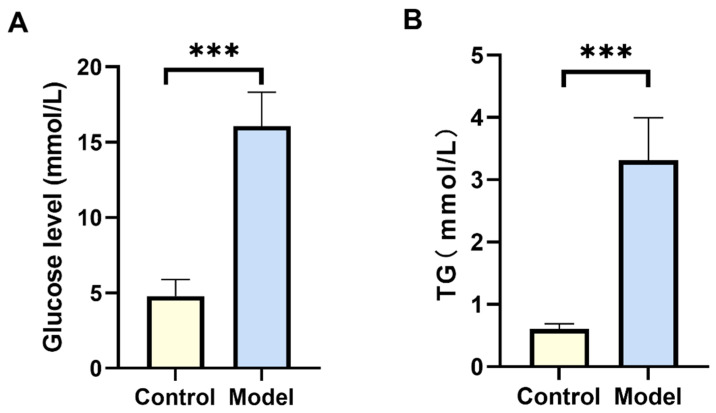
Blood glucose and lipid levels in silkworms. (**A**) Changes in silkworm glucose level after modeling. Significant differences between groups were evaluated using Student’s *t*-test. After 72 h of high-glucose diet feeding, silkworms in the model group had elevated blood glucose levels compared to the control group (*p* < 0.05). (**B**) Changes in silkworm TG level after modeling. TG levels in fat body lysate were detected using a TG kit. Significant differences between groups were evaluated using Student’s *t*-test. Compared with the control group, the TG content of the model group was high (*p* < 0.05). *** *p* < 0.001.

**Figure 2 biology-13-00557-f002:**
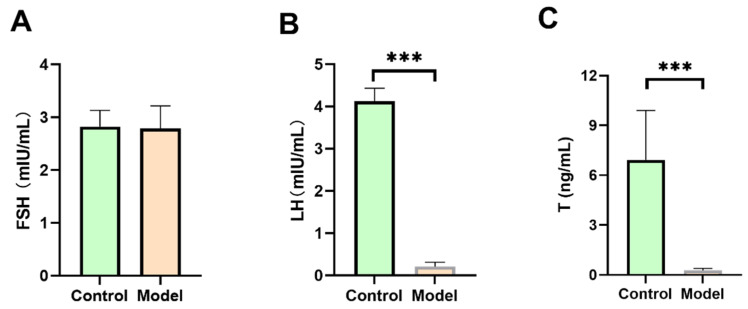
FSH, LH, and T levels in diabetic silkworms. The first prolegs of silkworms were cut to collect hemolymph. FSH, LH, and T levels in the hemolymph supernatant were detected using ELISA kits. (**A**) Effect of diabetes on FSH in silkworms. (**B**) Effect of diabetes on LH in silkworms. Compared with the control group, the LH level of the model group was low (*p* < 0.05). This result suggests that diabetes can lead to a decrease in silkworm LH content. (**C**) Effect of diabetes on T in silkworms. Compared with the control group, the LH level of the model group was low (*p* < 0.05). This result indicates that diabetes can lead to a decrease in silkworm T content. Significant differences between groups were evaluated using Student’s *t*-test. *** *p* < 0.001.

**Figure 3 biology-13-00557-f003:**
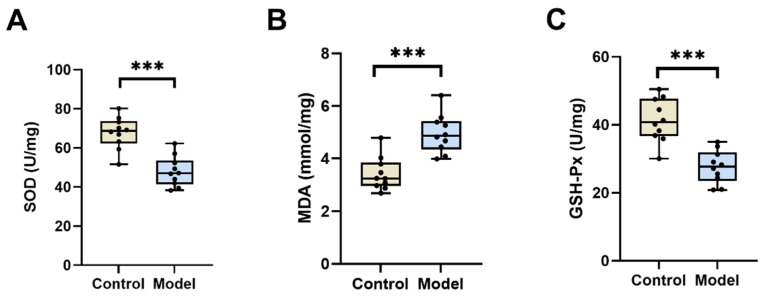
SOD, MDA, and GSH-Px levels in the testis of diabetic silkworms. SOD, MDA, and GSH-Px levels in silkworm testis were measured using kits. (**A**) SOD levels in testis of diabetic silkworms after modeling. The SOD level in the model group was significantly lower than in the control group (*p* < 0.05). (**B**) MDA levels in testis of diabetic silkworms after modeling. The MDA level in the model group was significantly higher than in the control group (*p* < 0.05). (**C**) GSH-Px levels in testis of diabetic silkworms after modeling. The GSH-Px level in the model group was significantly lower than in the control group (*p* < 0.05). Significant differences between groups were evaluated using Student’s *t*-test. *** *p* < 0.001.

**Figure 4 biology-13-00557-f004:**
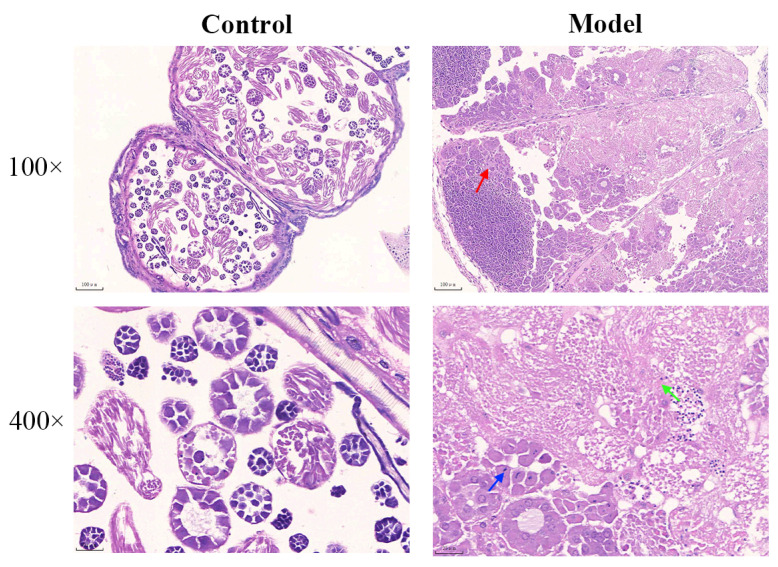
HE staining of silkworm testis. Silkworms in the control group exhibited an intact testicular cell structure, a high number of spermatogonia, and an intact seminal vesicle without any apparent pathological alterations. Silkworms in the model group displayed delayed testicular development, the necrosis of spermatogonia and seminal vesicles, and distinct pathological changes. The red arrows indicate the necrosis of spermatogonia, the blue arrow signifies the necrosis of seminal vesicles, and the green arrow indicates the exudation of plasma-fibrous material.

**Figure 5 biology-13-00557-f005:**
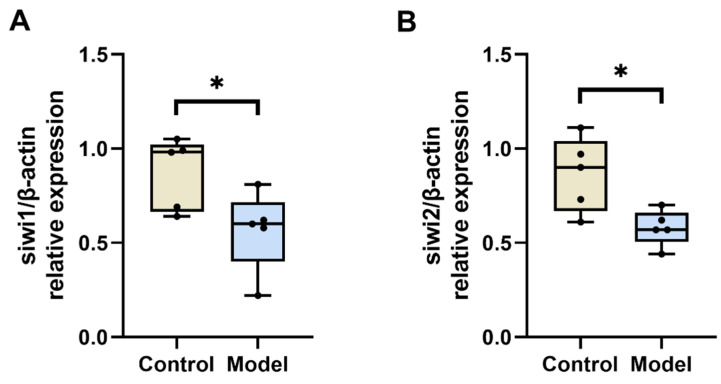
Expression of siwi1 and siwi2 in silkworm testis. Real-time PCR was used to detect the expression of siwi1 and siwi2 in silkworm testis. (**A**) Relative mRNA expression levels of siwi1 in silkworm testis. The relative expression of siwi1 in the model group was lower than in the control group (*p* < 0.05). (**B**) Relative mRNA expression levels of siwi2 in silkworm testis. The relative expression of siwi2 in the model group was lower than in the control group (*p* < 0.05). Significant differences between groups were evaluated using Student’s *t*-test. * *p* < 0.05.

**Table 1 biology-13-00557-t001:** PCR primer information.

Primer	Forward Primer	Reverse Primer
GAPDH	GCCGTGGTGCTCAACAAAACATC	ATGCCATTCCAGTCAGCTTGCC
siwi1	ACTCCCCTGGATCTGCTTGCC	TCCTCCGAGTGTTTTGCTGTGAAC
siwi2	GCACACGTCGTTTGAGCGAAAG	ATGTTGGCGTCCGAAGCGAATC

## Data Availability

All the datasets in this study can be provided upon reasonable request.

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
