# Peer review of "The Impact of Diabetes on Male Silkworm Reproductive Health"

_biology, 2024, doi:10.3390/biology13080557_

Round 1
Reviewer 1 Report
Comments and Suggestions for Authors
In this work, the authors demonstrate the economic feasibility of using silkworms as the model organism for the investigation of how diabetes affects male reproductive organs. The study is interesting and very important, especially in terms of using nontraditional model organisms.
1. The title and abstract capture the content and focus of the manuscript. However, the content of the abstract can be improved. For instance, the type of glucose fed to the silkworms should be mentioned before the result part (Lines 22-23).
2. The background and information provide an adequate understanding of the research yet can be improved. Since the study focuses on silkworms as a potential model, additional information on previous research on the use of silkworms for diabetic studies should be added to enrich Lines 64-71 in the introduction section.
3. The Material and Methods section can be improved. The write-up under this section should be systematic and concise. For instance, what is the name of the glucose diet used? What is the composition of the high-glucose diet? How was the feed prepared? Again, at what volume or amount was introduced into the silkworm, and what method of feeding was done (e.g., by injection or what)? What was the procedure involved in the measurement of glucose levels (sugar quantification)? How was damage to the reproductive organs measured or determined? Important details should be added in the "Experimental Model" section.
2. "2.2. Preparation of silkworm hemolymph" This section lacks the relevant details about the method performed.
3. An additional sub-section should be included for the purchase of reagents and other chemicals.
4. Each aspect of the Materials and Methods section should be described in a manner that is easily replicable by others. Descriptions that are vivid are required. Do not write a manuscript with the assumption that it will be read exclusively by experts!
5. Proper descriptions of figures that translate the results should be included. Figures are organized into subsections (a, b, c, etc.). As a result, it is imperative that authors specify the specific instances in the text where this is pertinent. For example, consider lines 129-130 (e.g., Figure 1a or 1b). The same procedure should be followed throughout the manuscript. Figures are poorly described in this current manuscript.
6. The resolutions of the figures throughout the manuscript are good. However, they are poorly described and lack adequate explanation. They are basic; the authors could have conducted an in-depth analysis for such a study.
7. The conclusion offers sufficient information regarding the potential of utilizing silkworms in the investigation of diabetes. Nevertheless, the male organ's response to the high-glucose diet is not documented. Therefore, it is in disregard of the manuscript's title. The authors exclusively concentrated on the viability of employing silkworms as model animals. Conclusion should be improved.
8. The references are appropriate
Author Response
Thank you very much for taking the time to review my manuscript. The main issues we've answered that in the "Response to Reviewer 1 Comments". Please see the attachment.

Reviewer 2 Report
Comments and Suggestions for Authors
This study innovatively uses silkworms as a model to investigate diabetic reproductive damage, highlighting cost-effectiveness, ethical advantages, and potential for drug screening. It provides valuable insights into the impacts of diabetes on reproductive health, emphasizing hormone levels and oxidative stress markers. However,the overall structure of the paper could be improved, with clearer separation and organization of sections to enhance readability and flow. There following concerns should be addressed
1: In the Abstract section, if the abbreviation is not mentioned after the abbreviation, please delete the brackets.
2. Line-43-45 please add the reference.
3. line-43-46 Some traditional herbal extracts are also used in the treatment of common diseases such as diabetes, please add this part to your list by referring to the following literature
#1 Yuan, H. D., Kim, J. T., Kim, S. H., & Chung, S. H. (2012). Ginseng and diabetes: the evidences from in vitro, animal and human studies. Journal of ginseng research, 36(1), 27.
#2 Chen W, Balan P, Popovich D G. Review of ginseng anti-diabetic studies[J]. Molecules, 2019, 24(24): 4501.
#3 Phong N V, Gao D, Kim J A, et al. Optimization of Ultrasonic-Assisted Extraction of α-Glucosidase Inhibitors from Dryopteris crassirhizoma Using Artificial Neural Network and Response Surface Methodology[J]. Metabolites, 2023, 13(4): 557.
4. The choice of glucose concentration and duration for inducing diabetes in silkworms is not well-justified or supported by references to previous studies.,please explain
5. 10 silkworms from each group is too limited
6. Line 93: why is 10 min? 15, 30,45 min is not suitable ?
7.line 116: H20 is not correct
8. Why only one item, elevated blood glucose, supports successful modeling.?
9. In the data analysis you used a t-test significance of 0.05, why did it change later?
10. Please clearly label the organizations in Figure 4
11. The study does not specify whether environmental factors were controlled for both the model and control groups, which could influence the results, please added.
12 The sample size is relatively small and not justified with a power analysis, which raises concerns about the statistical power of the study.
13 Details about the calibration and validation of the glucometer used for blood glucose measurements are missing, questioning the accuracy of these measurements.
14. The study does not provide information on the validation of ELISA kits used for hormone measurements, which could affect the reliability of the results.
15. The discussion section does not adequately compare the findings with those from similar studies on mammalian models, missing a chance to highlight the uniqueness or relevance of the study.
16 Ethical considerations are not discussed, even though the use of animal models, including invertebrates, typically warrants some discussion on ethical treatment, please added it
17. why not validate the gene expression results with an additional method, such as Western blotting, to confirm protein levels.
Comments on the Quality of English Language
This study innovatively uses silkworms as a model to investigate diabetic reproductive damage, highlighting cost-effectiveness, ethical advantages, and potential for drug screening. It provides valuable insights into the impacts of diabetes on reproductive health, emphasizing hormone levels and oxidative stress markers. However,the overall structure of the paper could be improved, with clearer separation and organization of sections to enhance readability and flow. There following concerns should be addressed
1: In the Abstract section, if the abbreviation is not mentioned after the abbreviation, please delete the brackets.
2. Line-43-45 please add the reference.
3. line-43-46 Some traditional herbal extracts are also used in the treatment of common diseases such as diabetes, please add this part to your list by referring to the following literature
#1 Yuan, H. D., Kim, J. T., Kim, S. H., & Chung, S. H. (2012). Ginseng and diabetes: the evidences from in vitro, animal and human studies. Journal of ginseng research, 36(1), 27.
#2 Chen W, Balan P, Popovich D G. Review of ginseng anti-diabetic studies[J]. Molecules, 2019, 24(24): 4501.
#3 Phong N V, Gao D, Kim J A, et al. Optimization of Ultrasonic-Assisted Extraction of α-Glucosidase Inhibitors from Dryopteris crassirhizoma Using Artificial Neural Network and Response Surface Methodology[J]. Metabolites, 2023, 13(4): 557.
4. The choice of glucose concentration and duration for inducing diabetes in silkworms is not well-justified or supported by references to previous studies.,please explain
5. 10 silkworms from each group is too limited
6. Line 93: why is 10 min? 15, 30,45 min is not suitable ?
7.line 116: H20 is not correct
8. Why only one item, elevated blood glucose, supports successful modeling.?
9. In the data analysis you used a t-test significance of 0.05, why did it change later?
10. Please clearly label the organizations in Figure 4
11. The study does not specify whether environmental factors were controlled for both the model and control groups, which could influence the results, please added.
12 The sample size is relatively small and not justified with a power analysis, which raises concerns about the statistical power of the study.
13 Details about the calibration and validation of the glucometer used for blood glucose measurements are missing, questioning the accuracy of these measurements.
14. The study does not provide information on the validation of ELISA kits used for hormone measurements, which could affect the reliability of the results.
15. The discussion section does not adequately compare the findings with those from similar studies on mammalian models, missing a chance to highlight the uniqueness or relevance of the study.
16 Ethical considerations are not discussed, even though the use of animal models, including invertebrates, typically warrants some discussion on ethical treatment, please added it
17. why not validate the gene expression results with an additional method, such as Western blotting, to confirm protein levels.
Author Response
Thank you very much for taking the time to review my manuscript. The main issues we've answered that in the "Response to Reviewer 2 Comments". Please see the attachment.

Round 2
Reviewer 2 Report
Comments and Suggestions for Authors
The current form can be accepted
Comments on the Quality of English LanguageQuality of English Language is fine